METHODS AND RESOURCES

# A curated data resource of 214K metagenomes for characterization of the global antimicrobial resistome

**Hannah-Marie Martiny** *, **Patrick Munk**, **Christian Brinch**, **Frank M. Aarestrup**, **Thomas N. Petersen**

Research Group for Genomic Epidemiology, Technical University of Denmark, Kongens Lyngby, Denmark

* hanmar@food.dtu.dk

## Abstract

The growing threat of antimicrobial resistance (AMR) calls for new epidemiological surveillance methods, as well as a deeper understanding of how antimicrobial resistance genes (ARGs) have been transmitted around the world. The large pool of sequencing data available in public repositories provides an excellent resource for monitoring the temporal and spatial dissemination of AMR in different ecological settings. However, only a limited number of research groups globally have the computational resources to analyze such data. We retrieved 442 Tbp of sequencing reads from 214,095 metagenomic samples from the European Nucleotide Archive (ENA) and aligned them using a uniform approach against ARGs and 16S/18S rRNA genes. Here, we present the results of this extensive computational analysis and share the counts of reads aligned. Over $6.76 \cdot 10^8$ read fragments were assigned to ARGs and $3.21 \cdot 10^9$ to rRNA genes, where we observed distinct differences in both the abundance of ARGs and the link between microbiome and resistome compositions across various sampling types. This collection is another step towards establishing global surveillance of AMR and can serve as a resource for further research into the environmental spread and dynamic changes of ARGs.

## Introduction

The vast amount of genomic data available in public data repositories is a unique and potentially important resource for doing research and genomic surveillance of antimicrobial resistance (AMR). Using these datasets collected from locations all over the world across different years and from various sampling sources might further aid our understanding of the emergence and distribution of antimicrobial resistance genes (ARGs).

The sharing of genomic sequence data to one of the available repositories is today a major and often mandatory step in peer-reviewed journals, for which several repositories were created by the members of the International Nucleotide Sequence Database Collaboration (INSDC) [1], including the European Nucleotide Archive (ENA) [2]. The number of sequencing data available at ENA continues to increase with an estimated doubling time of 18 months (https://www.ebi.ac.uk/ena/browser/about/statistics; accessed 2022-03-08).

**Data Availability Statement:** The code to produce the figures is available at https://github.com/hmmartiny/mARG. The data has been deposited at https://doi.org/10.5281/zenodo.6919377, and

documentation of the various tables can be accessed at https://hmmartiny.github.io/mARG.

**Funding:** This work was supported by the European Union's Horizon H2020 grant VEO (874735) and the Novo Nordisk Foundation (grant NNF16OC0021856: Global Surveillance of Antimicrobial Resistance). HMM, PM, CB, TNP, and FMA were all supported by both grants. The funders had no role in study design, data collection and analysis, decision to publish, or preparation of the manuscript.

**Competing interests:** The authors have declared that no competing interests exist.

**Abbreviations:** AMR, antimicrobial resistance; ARG, antimicrobial resistance gene; ENA, European Nucleotide Archive; INSDC, International Nucleotide Sequence Database Collaboration; mcr, mobilized colistin resistance.

Several approaches for analyzing genomic data depending on the sample types are already well established.

However, the exploration of these resources is often restricted to a few research groups only since both sufficient skills in bioinformatics and access to high-performing computer resources are needed to handle the large amount of available data.

Existing collections of analyzed datasets tend to focus on either specific sample sources, such as humans [3,4], marine [5], or urban sewage [6,7], or focus on specific genera [8]. Especially the COVID-19 pandemic has highlighted the value of data sharing to trace the spread and evolution of the virus [9]. Despite the attempts to standardize the analysis workflows of these databases, they are limited in their ability to generalize across environments and locations. A recent study [10] has shared a searchable collection of 661K bacterial genomes for exploring the global bacterial diversity across different origins, providing an easy-to-access resource for genomic research. While this is an impressive data-sharing effort, the authors did not include metagenomic samples in their pipeline. Metagenomic techniques aim to sequence all DNA in a sample and can be used to characterize the microbiome in different environments [11,12], discover novel organisms [13], monitor disease [14,15], and specific genes, such as ARGs [5,6,16].

Here, we present a large-scale metagenomic analysis of 214,095 metagenomic samples retrieved from ENA. We have carried out an assembly-free approach by aligning sequencing reads against ARGs and 16S/18S ribosomal RNA genes. We have previously published an in-depth analysis of the distribution of mobilized colistin resistance [17] based on those data. Now we both share the entire collection of mapping results and showcase how to characterize the global resistome and microbiome with this dataset. The curated metadata and mapping results are available at https://doi.org/10.5281/zenodo.6919377 and documentation at https://hmmartiny.github.io/mARG/Tables.html.

## Materials and methods

### Retrieval of metagenomes

We retrieved metagenomic datasets from ENA [2] uploaded between 2010-01-01 and 2020-01-01 that had library source as "METAGENOMIC" and library strategy of "WGS." We collected 214,095 sequencing runs from 146,732 samples from 6,307 projects corresponding to 442 Tbp of raw reads taking up 300 TB of storage. The associated metadata for each sample was also retrieved.

### Preprocessing and mapping of sequencing reads

The retrieved raw FASTQ reads were trimmed and aligned against reference sequences, as outlined in Martiny (2022) [17]. In brief, we used FASTQC v.0.11.15 (https://www.bioinformatics.babraham.ac.uk/projects/fastqc/) for read quality checking and BBduk2 v.36.49 [18] for trimming the raw sequencing reads. With the k-mer-based alignment tool KMA 1.2.21 [19], the trimmed reads were mapped against reference sequences from 2 different databases: The AMR gene database ResFinder [20] (downloaded 2020-01-25), which contained 3,085 sequences of acquired ARGs, and the ribosomal rRNA Silva [21] gene database (version 138, downloaded 2020-01-16), which had 2,225,272 reference sequences with more than 88% of them being 16/18S rRNA genes. For KMA, we used the following alignment parameters: 1, -2, -3, -1 for a match, mismatch, gap opening, and gap extension. For read pairing, we used a value of 7 and a minimum relative alignment score of 0.75. Data retrieval, quality checking, trimming, and read alignments were done using the Danish National Supercomputer for Life Sciences (https://www.computerome.dk/).

## Standardization of metadata

The following attributes for each metagenome were standardized: sampling location, sampling host or environment (referred to as a host below), and sampling date.

To standardize the label for sampling locations, we looked at the values entered in the two fields "country" and "location." First, the latitude and longitude coordinates were mapped to a country using the Python library Shapely 1.7.1 [22] to find the matching area defined in one of the 3 public domain map datasets (countries, marine, and lakes) available in the Natural Earth Data collection. If the lookup failed or the coordinates were not given, the second step was to match the text attribute in the country label to ISO 3166 country codes with a fuzzy search with the Python library PyCountry 20.7.3 (https://github.com/flyingcircusio/pycountry). Finally, if the 2 lookup searches did not yield a match, we did a manual lookup of the country labels to standardize the text.

For the standardization of host labels, we mapped the taxonomic id given by the attribute "host_tax_id" to the NCBI Taxonomy database [23], or if the feature was missing, the "tax_id" was used instead.

Since the only way to curate entered collection dates is to look up suspicious dates in published studies manually, and that was deemed too time-intensive, we decided to replace dates entered as later than 2020-01-01 in the sample attribute field "collection_date" with the missing value NULL.

## Measuring the abundance of ARGs

Since we report the fragment count aligned to each reference gene, the mapping results are compositional and should be treated as such [24]. In the simplest form, the ARG abundance for a sample or sample group can be calculated as the log-ratio of the count of reads, $n_i$, aligned to each ARG $i$ over the total sum of rRNA read fragments $n_B$:

$$x = [n_1, n_2, \ldots, n_D, n_B], i = 1..D$$

$$\text{Abundance}(x) = \left[ \log \frac{n_1}{n_B}, \log \frac{n_2}{n_B}, \ldots, \log \frac{n_D}{n_B} \right]$$

where $D$ is the number of ARGs and $n_B = \frac{\sum_j^{D_B} n_j}{1 \cdot 10^6}$ with $D_B$ being the number of read fragments aligned to rRNA genes. Each ARG count $n_i$ has been adjusted with the length of the gene in kilobases.

The relative abundance resistance classes were calculated as the proportion of ARG resistance assigned to different classes and scaled with $\kappa = 100$:

$$\text{Relative abundance}(x) = \frac{\kappa}{\sum n_i} n_i$$

## Diversity measurements

Besides the read abundance values, we report the species richness, Shannon diversity index [25], and the Gini–Simpson [26] diversity index of read counts of ARGs, genera, and phyla per sample. Species richness is the number of different genes or taxonomic groups present in the sample with at least 1 read fragment aligned.

The Shannon index ($H'$) was calculated using the proportions of reads $p_i = \frac{n_i}{\sum n}$:

$$H' = -\sum_{i=1}^{R} p_i \ln p_i$$

whereas the Gini–Simpson index ($GS$) was calculated using the read counts $n = [n_1, \ldots, n_D]$ and $N = \sum n$ is the total count of reads for the group:

$$GS = 1 - \frac{\sum n_i \cdot (n_i - 1)}{N \cdot (N - 1)}$$

Together with these 2 indices, we also report the sample-wise unique number of reference sequences or taxonomic groups matched.

## Results

Here, we present a large-scale mapping of 442 Tbp of raw reads of 214,095 metagenomic samples suitable for analyzing the distribution of acquired antimicrobial resistance genes and 16S/18S rRNA genes. Furthermore, we have spent considerable effort standardizing 3 main sample attributes: sampling date, location, and source. To facilitate easy access and usage, we have shared the mapping results and corrected metadata in 3 different data formats (TSV, HDF, and MySQL dumps). We also provide tutorials with code examples in R and Python on using the data in different scenarios. Data files are all available at https://doi.org/10.5281/zenodo.6919377.

By collecting the sequencing reads from ENA, we could also verify the inherited bias of specific sample types or sources being overrepresented simply due to the availability in the public repository. While the 214,095 metagenomic datasets were collected from 797 different hosts, most were either of human or marine origin (Fig 1A). A similar skewed geographical distribution towards European and North American countries was observed in the sampling locations

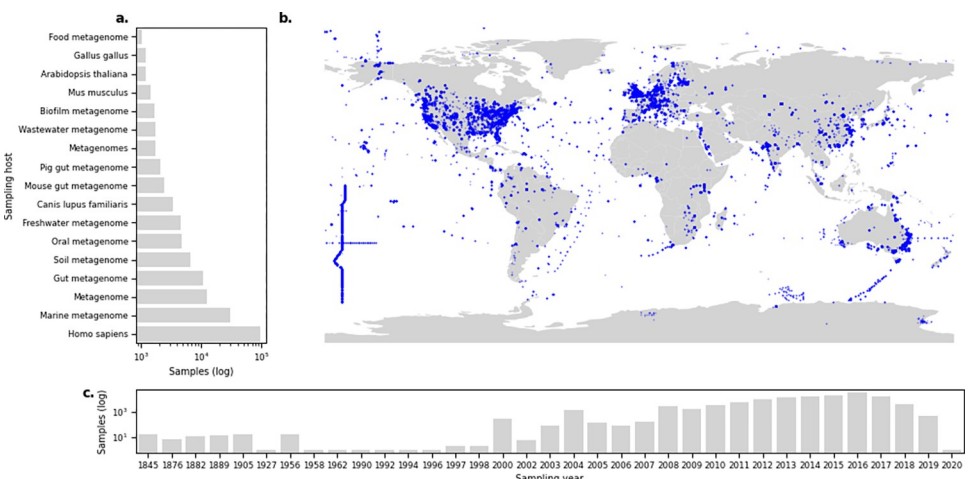

**Fig 1. Distribution of metagenomes reveals the overrepresentation of samples from specific sources. (a)** Number of samples grouped per sampling host, where only hosts with more than 1,000 samples are plotted. (**b**) Sample locations for metagenomes with available GPS coordinates; each marker is a sample. A total of 83,903 samples did not have coordinates available. (**c**) Year of which a sample was collected. A total of 84,238 of the samples did not have a valid sampling date recorded. The data underlying this figure can be found at https://doi.org/10.5281/zenodo.6919377, and the base layer map was created with data from https://www.naturalearthdata.com/.

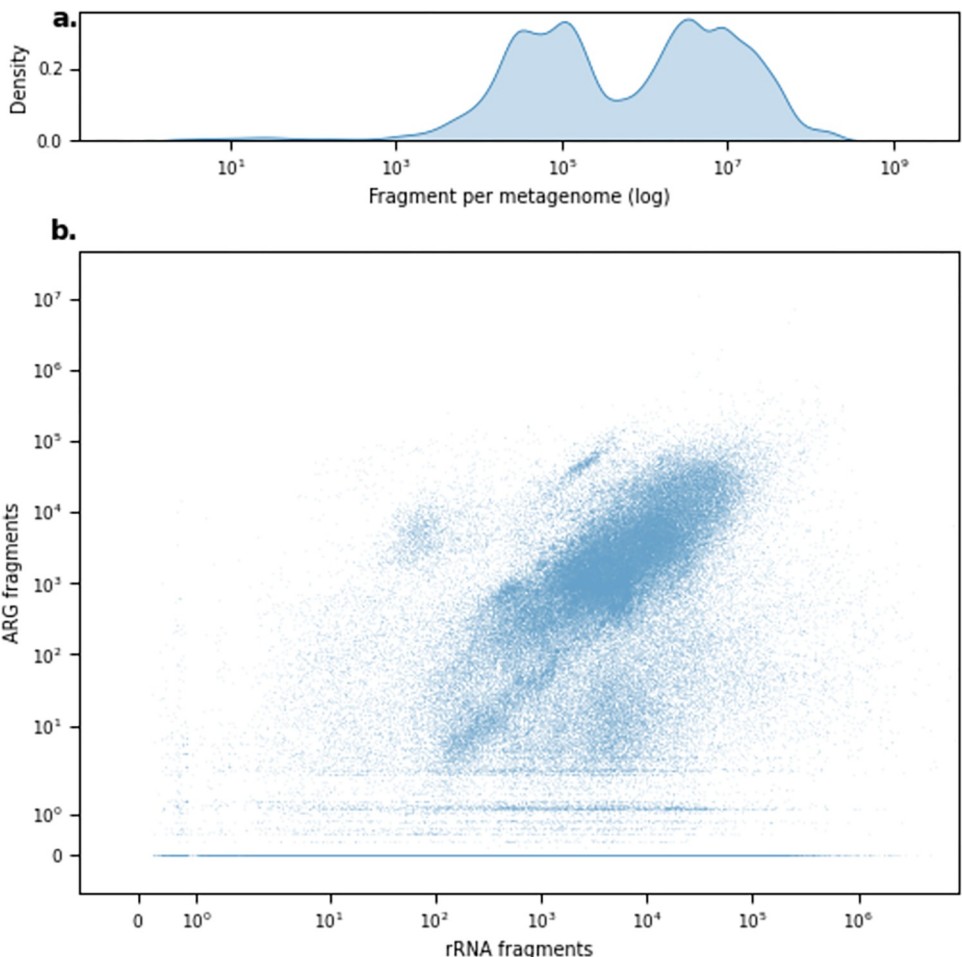

**Fig 2. Distribution of available and aligned fragments. (a)** Density distribution of available fragments per sample.
(**b**) The distribution compares the number of fragments mapped to rRNA genes and ARGs. The data underlying this
figure can be found at https://doi.org/10.5281/zenodo.6919377.

(Fig 1B). The distribution of samples according to the sampling year reveals that a considerable
number were collected between 2010 and 2020 (Fig 1C).

Of the more than $1.8\cdot10^{12}$ raw sequencing reads, corresponding to 442.1 Tbp, 93% of the
reads were generated using Illumina sequencing technologies (S1 Fig). We mapped over
$1.69\cdot10^{12}$ trimmed read fragments, with a median of 784,748 fragments per sample (range 1 to
916,901,400) (Fig 2A). Approximately 0.04% of all read fragments could be aligned to ARGs,
and 0.19% to rRNA genes. Overall, the amount of sequencing reads and bases available did
increase the count of aligned read fragments (S3 Fig). The number of ARG fragments aligned
increased with the number of aligned rRNA fragments, although for 34% of the samples, we
did not find any ARGs despite having read fragments aligning to 16S rRNA genes (Fig 2B).
The microbial differences in the different sampling origins were highlighted in the number of
aligned fragments (S4 Fig).

## The global abundance of antimicrobial resistance

To measure the global distribution of ARGs and the composition of the resistome, we calcu-
lated the abundance of ARGs as the log-ratio of ARG fragments over summed rRNA sequence

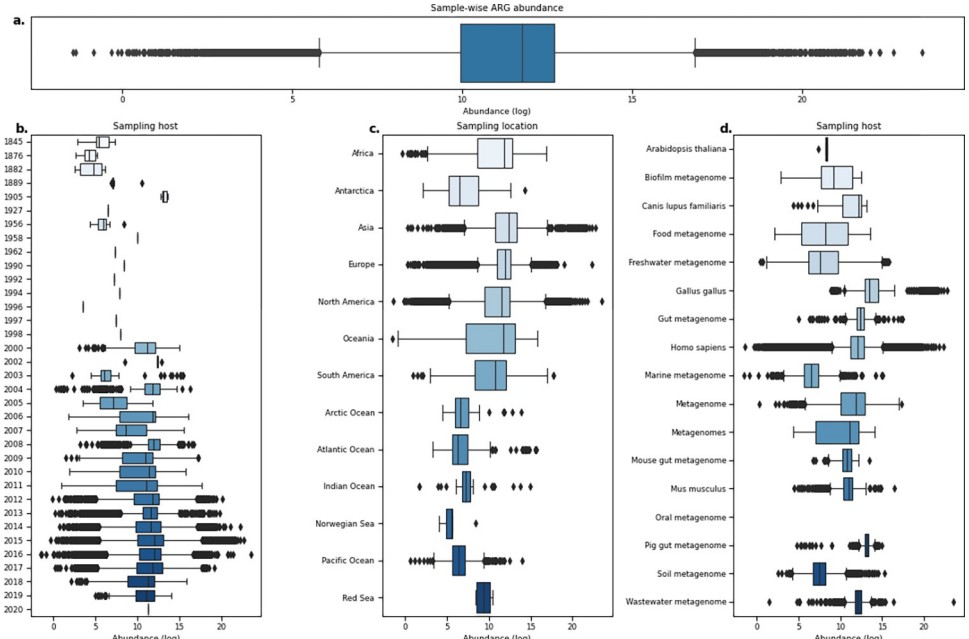

**Fig 3. Boxplots of ARG abundances in metagenomic samples show that levels vary across different origins. (a)**
Distribution of ARG abundance per sample. **(b)** Distribution of sample-wise ARG abundance grouped by sampling
year. **(c)** Sample-wise ARG abundance per sampling location. **(d)** Sample-wise ARG abundance grouped by hosts.
Only hosts with more than 1,000 metagenomes analyzed are shown. The data underlying this figure can be found at
https://doi.org/10.5281/zenodo.6919377.

fragments. Almost all of the reference sequences from the ResFinder database had at least 1
fragment aligned, and only 94 ARGs had no hits (S2 Fig). The median observed resistance load
per metagenomic sample was 11.74 (log range: −1.45 to 23.52) (Fig 3A), which appeared to be
mainly dependent on the geographic origin and environment (Fig 3B–3D) and not on which
year the sample was taken. For example, samples originating from locations within Europe
showed similar abundance levels for most of the samples but with several outliers, whereas
multiple samples from locations in the Oceania region had a much broader load distribution
with few outliers (Fig 3C).

While the distribution of sample-wise resistance loads illustrates the high variability in this
data collection (Fig 3), we saw that once we stratified the relative ARG read proportions per
resistance class and sample type, there were clear separations between different groups (Fig 4).
For the sampling years with a considerable number of samples available (2004 to 2019), the rel-
ative proportion of classes was relatively consistent, with Tetracycline reads being the most
common, except for a spike of Beta-lactam reads in 2017 (Fig 4A). Across the continents and
large water bodies, we observed that ARGs conferring resistance to Aminoglycosides or Beta-
lactam antimicrobials were more common in water environments, whereas mainland regions
had a more diverse distribution (Fig 4B). Once we stratified by sampling host or source, the
distribution of resistance classes was very dependent on the group, as seen by the high propor-
tion of read fragments aligned to, for example, Phenicol for marine and soil samples and Tetra-
cycline reads being highly prevalent in mice (*Mus musculus*) samples (Fig 4C).

## Linking the microbiome diversity with resistance diversity

The relationship between the diversity of the microbiome and the resistance genes was quanti-
fied by calculating the species richness and 2 alpha diversity measurements (Shannon and

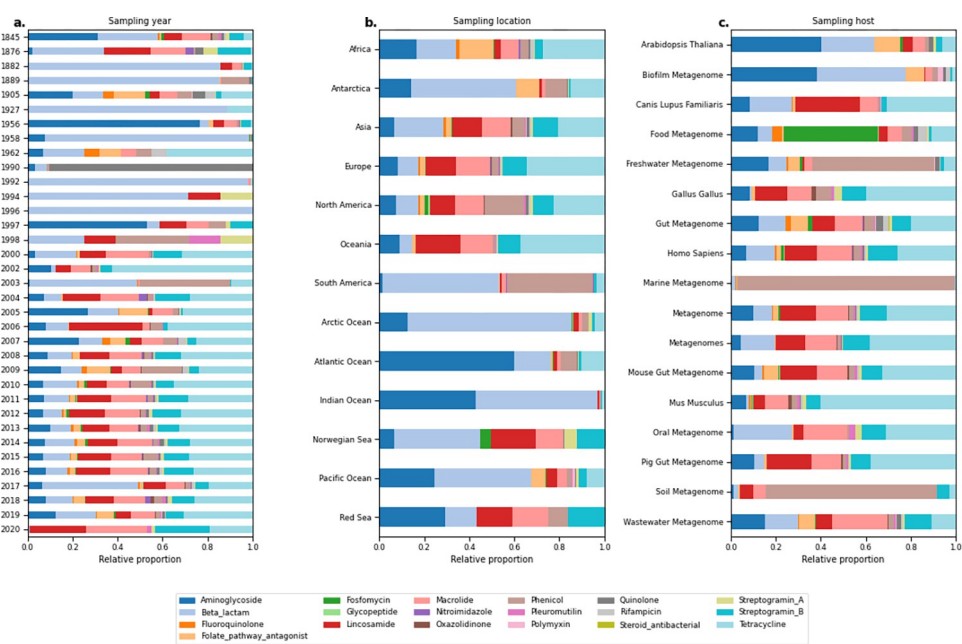

**Fig 4. Composition of reads assigned to ARGs from different resistance classes grouped by sampling origin. (a)** Grouped by sampling year. (**b**) Grouped per sampling location. (**c**) Grouped per sampling host. Only hosts with more than 1,000 metagenomes analyzed are shown. The data underlying this figure can be found at https://doi.org/10.5281/zenodo.6919377.

Gini–Simpson) on ARG levels and phyla and genera taxonomic levels. Without looking at the sample origin, we observed that a majority of the samples had both high microbial diversity and ARG diversity (Figs 5 and S5). However, the relationship between genera and ARG diversity indexes differed between sampling sources, with several groups containing samples that

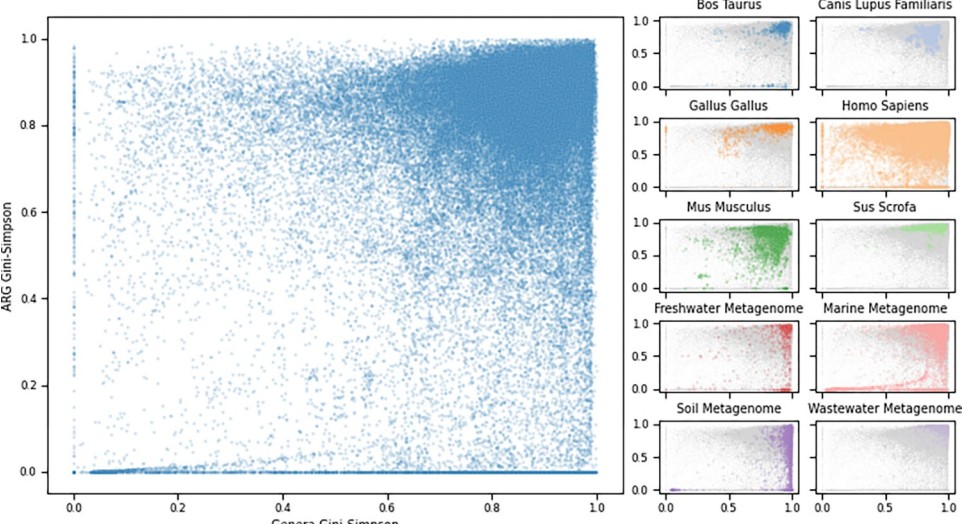

**Fig 5. The genus–ARG diversity relationship for all metagenomic samples.** The Gini–Simpson diversity indexes were calculated on genus categories (x-axis) compared to ARG levels (y-axis). Left: scatterplot of all samples. Right: samples colored by selected host or environmental origins. The data underlying this figure can be found at https://doi.org/10.5281/zenodo.6919377.

did not follow the assumption of the 2 diversity measurements following each other, suggesting that increased diversity of microbes in, for example, soil samples does not necessarily lead to a higher diversity of resistance genes. Contrarily, the chicken (*Gallus gallus*) samples showed that they still had elevated ARG diversity despite having lower microbial diversity (Fig 5).

## Discussion

Global surveillance of AMR based on genomics continues to become more accessible due to the advancement in NGS technologies and the practice of sharing raw sequencing data in public repositories. Standardized pipelines and databases are needed to utilize these large data volumes for tracking the dissemination of AMR. We have uniformly processed the sequencing reads of 214,095 metagenomes for the abundance analysis of ARGs.

Our data sharing efforts enable users to perform abundance analyses of individual ARGs, the resistome, and the microbiome across different environments, geographic locations, and sampling years.

We have given a brief characterization of the distribution of ARGs according to the collection of metagenomes. However, in-depth analyses remain to be performed to investigate the influence of temporal, geographical, and environmental origins on the dissemination and evolution of antimicrobial resistance. For example, analyzing the spread of specific ARGs across locations and different environments could reveal new transmission routes of resistance and guide the design of intervention strategies to stop the spread. We have previously published a study focusing on the distribution of mobilized colistin resistance (*mcr*) genes using this data resource, showing how widely disseminated the genes were [17]. Another use of the data collection could be to explore how the changes in microbial abundances affect and are affected by the resistome. Furthermore, our coverage statistics of reads aligned to ARGs could be used to investigate the rate of new variants occurring in different reservoirs. Even though we have focused on the threat of antimicrobial resistance, potential applications of this resource can be to look at the effects of, for example, climate changes on microbial compositions. Linking our observed read fragment counts with other types of genomic data, such as evaluating the risk of ARG mobility, accessibility, and pathogenicity in assembled genomes [27,28], and verifying observations from clinical data [29].

We recommend that potential users consider all the confounders present in this data collection in their statistical tests and modeling workflows, emphasizing that the experimental methods and sequencing platforms dictate the obtained sequencing reads and that metadata for a sample might be mislabeled, despite our efforts to minimize those kinds of errors. Furthermore, it is essential to consider the compositional nature of microbiomes [30]. The reads do not depend on the distribution of genetic material in the sample but on the capacity of the sequencing platform [24,31]. Various statistical methods already exist that consider the compositionality [24,32,33]. Finally, it is important to highlight that the results we have presented here include fragment counts of 1 for the sake of transparency, but we also recommend potential users consider appropriate filters in their analysis.

The sequencing data in public repositories has continued to grow, giving us plenty of opportunities to continue to expand our data collection even more. To establish a truly global surveillance program of AMR, sequencing data should be analyzed as soon as published in these archives. Although this would require access to even more computational resources, we hope to achieve this soon and compare our approach with other methods, such as AMRFinderPlus [34] and CARD [35]. As new sequencing technologies are becoming more used, our settings for our alignment procedure should also be tuned to better take advantage and be aware of the flaws of different sequencing platforms.

With this data resource, we have taken a step towards enabling the scientific community to utilize the wealth of information in these metagenomic samples to broaden our understanding of the dissemination of antimicrobial resistance and changes in microbiomes at both local and global scales through time and environments.

## Supporting information

**S1 Fig. Distribution of samples per sequencing instrument platform. (a)** Sample count per platform. (**b**) Distribution of raw sequencing read counts per platform. The data underlying this figure can be found at https://doi.org/10.5281/zenodo.6919377.
(TIFF)

**S2 Fig. More than 96% of ARG templates had at least 1 aligned fragment.** The bars illustrate the percentage of ARGs per resistance class without and with at least 1 aligned fragment. The parenthesis after each class label contains the number of genes found out of the total available templates. The data underlying this figure can be found at https://doi.org/10.5281/zenodo.6919377.
(TIFF)

**S3 Fig.  The sample-wise distribution of aligned (a) ARG or (b) rRNA fragments compared to raw sequencing base counts.** The data underlying this figure can be found at https://doi.org/10.5281/zenodo.6919377.
(TIFF)

**S4 Fig. The sample-wise distribution of aligned rRNA fragments and ARG fragments, colored by selected host and environmental sources.** The data underlying this figure can be found at https://doi.org/10.5281/zenodo.6919377.
(TIFF)

**S5 Fig. Additional distributions showing the relationship between ARGs and genera for all metagenomic samples. (a)** The richness of genus groups (x-axis) vs. ARG richness (y-axis). (**b**) The relationship between Shannon diversity index calculated on genus level (x-axis) and ARGs (y-axis). Right: samples colored by selected host or environmental origins. The data underlying this figure can be found at https://doi.org/10.5281/zenodo.6919377.
(TIFF)

## Author Contributions

**Conceptualization:** Frank M. Aarestrup, Thomas N. Petersen.

**Data curation:** Hannah-Marie Martiny.

**Formal analysis:** Hannah-Marie Martiny.

**Funding acquisition:** Frank M. Aarestrup, Thomas N. Petersen.

**Investigation:** Hannah-Marie Martiny, Patrick Munk.

**Methodology:** Hannah-Marie Martiny, Patrick Munk, Thomas N. Petersen.

**Project administration:** Patrick Munk, Frank M. Aarestrup, Thomas N. Petersen.

**Resources:** Hannah-Marie Martiny, Frank M. Aarestrup.

**Software:** Hannah-Marie Martiny.

**Supervision:** Patrick Munk, Christian Brinch, Frank M. Aarestrup, Thomas N. Petersen.

**Validation:** Hannah-Marie Martiny, Christian Brinch.

**Visualization:** Hannah-Marie Martiny.

**Writing – original draft:** Hannah-Marie Martiny.

**Writing – review & editing:** Hannah-Marie Martiny, Patrick Munk, Christian Brinch, Frank M. Aarestrup, Thomas N. Petersen.

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
