## [Editor Report · Decision Letter 0]

19 May 2022

Dear Dr Martiny, 

Thank you for submitting your manuscript entitled "A curated data resource of 214K metagenomes for characterization of the global resistome" for consideration as a Methods and Resources by PLOS Biology.

Your manuscript has now been evaluated by the PLOS Biology editorial staff, as well as by an academic editor with relevant expertise, and I'm writing to let you know that we would like to send your submission out for external peer review.

Once your full submission is complete, your paper will undergo a series of checks in preparation for peer review. Once your manuscript has passed the checks it will be sent out for review. To provide the metadata for your submission, please Login to Editorial Manager (https://www.editorialmanager.com/pbiology) within two working days, i.e. by May 23 2022 11:59PM.

If your manuscript has been previously reviewed at another journal, PLOS Biology is willing to work with those reviews in order to avoid re-starting the process. Submission of the previous reviews is entirely optional and our ability to use them effectively will depend on the willingness of the previous journal to confirm the content of the reports and share the reviewer identities. Please note that we reserve the right to invite additional reviewers if we consider that additional/independent reviewers are needed, although we aim to avoid this as far as possible. In our experience, working with previous reviews does save time. 

If you would like to send previous reviewer reports to us, please email me at rroberts@plos.org to let me know, including the name of the previous journal and the manuscript ID the study was given, as well as attaching a point-by-point response to reviewers that details how you have or plan to address the reviewers' concerns. 

Kind regards,

Roli Roberts

Roland Roberts

Senior Editor

PLOS Biology

rroberts@plos.org

---

## [Decision Letter · Decision Letter 1]

14 Jul 2022

Dear Dr Martiny,

Thank you for your patience while your manuscript "A curated data resource of 214K metagenomes for characterization of the global resistome" was peer-reviewed at PLOS Biology. It has now been evaluated by the PLOS Biology editors, an Academic Editor with relevant expertise, and by three independent reviewers. 

Based on the broadly very favourable reviews, we are likely to accept this manuscript for publication, provided you satisfactorily address the points raised by the reviewers. Please also make sure to address the following data and other policy-related requests.

a) Please address the concerns raised by the three reviewers.

b) Please could you change your Title to something slightly more explicit for our wider readership? We suggest "A curated data resource of 214K metagenomes characterizes the global antimicrobial resistome"

c) Please address my Data Policy requests below; specifically, we need you to supply the numerical values underlying Figs 1ABC, 2AB, 3ABCD, 4ABC, 5, S1AB, S2, S3, S4AB. I note that your Zenodo deposition currently only seems to contain relatively “raw” values, rather than those directly shown in the Figure – please could you include the latter, clearly labelled? If you’ve used any custom code, please also include this.

d) Please also cite the location of the data clearly in each main and supplementary Fig legend, e.g. “The data underlying this Figure can be found in https://doi.org/10.5281/zenodo.6519844”

We expect to receive your revised manuscript within two weeks. 

*Published Peer Review History*

*Press*

Sincerely,

Roli Roberts

Roland Roberts, PhD

Senior Editor,

rroberts@plos.org,

PLOS Biology

DATA POLICY:

Regardless of the method selected, please ensure that you provide the individual numerical values that underlie the summary data displayed in the following figure panels as they are essential for readers to assess your analysis and to reproduce it: Figs 1ABC, 2AB, 3ABCD, 4ABC, 5, S1AB, S2, S3, S4AB. NOTE: the numerical data provided should include all replicates AND the way in which the plotted mean and errors were derived (it should not present only the mean/average values).

DATA NOT SHOWN?

REVIEWERS' COMMENTS:

Reviewer #1:

Martiny et al. describe a new data resource that is the product of intensive and large-scale bioinformatics analysis of metagenomic data for the presence and abundance of acquired antimicrobial resistance genes (ARGs). This paper can be viewed from two perspectives: (1) a data science contribution to allow the community to better examine AMR transmission patterns and (2) knowledge gained from analysis of the data. 

DATA SCIENCE

From the data science perspective, this is a very significant contribution to the field using the latest standards and mining of >200,000 metagenomic datasets, totalling more than 400 TB of sequence data. Considerable effort was put into data harmonization and normalization to provide a high-value data set to AMR researchers. The data, pipelines, software, and results are provided in a well-organized open format, allowing their analysis by the broader community. As the amount of computation needed to produce this data set is well beyond most AMR researchers interested in using genomics to understand ARG transmission patterns, this contribution is novel and of high value. 

Software and their versions are properly described in the methods section, but parameters used for KMA are not outlined. Were default parameters used? It is fine that the methods are presented "in brief" given a citation of previous work, but the manuscript would be improved if this included the cut-offs used by KMA to determine aligned reads (MAPQ?). Similarly, a more explicit statement in the methods that use of ResFinder focuses on the analysis of acquired ARGs and does not include resistance via mutation (e.g. PointFinder) would be helpful.

ANALYSIS

Analysis and interpretation of the data are thin, with some issues that need to be addressed, but this does not undermine that the primary purpose of the manuscript is to describe the generation and content of the data produced for the broader community. Full analyses of these data are beyond the scope of this manuscript and the authors perform an adequate overview analysis and summary of the major sub-sets and trends in the data. However, the statement of "a general trend" in lines 227-229 does not appear supported by Figures 5 & S4. This section should be re-written to carefully discuss patterns supported by the data, such as exists for the chicken data, instead of broad statements based on unconvincing patterns in the plots.

The data include ARGs that have as little as a single read fragment aligned and these ARGs were used in the species richness estimates. Can the authors explain why they did not include a minimum coverage cut-off in these analyses? 

The results presented are broken down by host (i.e. environment), location, and ResFinder drug class, but not by ARG families. While others are very likely to analyze these data for transmission patterns of ARGs or ARG families, at least an anecdotal investigation of a few ARGs would help illustrate the value of the data. Perhaps something recent like MCR versus the AACs? The "trends" mentioned above may be more obvious at the level ARG families.

DISCUSSION

Successful annotation of metagenomics data for ARGS requires both good software for sequencing read alignment and good reference data. Both KMA and ResFinder reflect the latest standards but like CARD and other databases, ResFinder's reference data is primarily from clinical isolates. It is possible there are ARGs in the environmental metagenomics data that are sufficiently different from these reference data to a degree that KMA is unsuccessful. CARD has its "Resistomes & Variants" data to provide an alternate in silico diversity of >200,000 ARG alleles for sequence read alignment. I'm not suggesting a re-analysis of these data with a broader in silico reference sequence collection, but I think the discussion should address this possible bias, i.e. false-negative results for divergent ARGs because of the algorithm/reference choice.

As mentioned above, the manuscript has little in the assessment of ARG transmission patterns, which is fine as it was not the major point of the paper, but Zhang et al. (PMID 34362925; Nature Communications 12: 4765) & Zhang et al. (PMID 35322038; Nature Communications 13: 1553) have recently published some large scale ARG metagenomic analyses that included assessment of ARG transmission and generation of risk metrics. At a minimum, the discussion should place the author's work in the context of these recent efforts. 

As mentioned above, the data include ARGs that have as little as a single read fragment aligned. The authors should add a statement that they are including these data for complete transparency so others can decide their own cut-offs when analyzing the data.

No information is given in the discussion on the long-term maintenance of this resource. What is the plan as new metagenomics data become available? CARD has (beta) pathogen-of-origin kmer tools for ARGs, will the authors be exploring similar methods to provide a more pathogen-centric perspective in future analyses?

MINOR POINTS

Figure 1C caption should mention the number of samples for which the collection date was NULL.

Lines 210-214 have very confusing grammar.

The phrase "ARG template" is used without proper definition.

Reviewer #2:

The resource presented by Martiny et al. is timely and has potential to boost the research antimicrobial resistance through widening access. The methods are presented clearly, and the datasets are made publicly available. There are a few points that I recommend the authors to consider towards improving the quality through cross-checking some of the analyses.

1. Given the impressive volume and the breadth of the data analysed, it is quite surprising that 96 ARGs did not have any alignments. A cross-cheling of these results and any indicators of the underlying reasons (e.g., these ARGS being very specific to the environments not being represented here?) will be important.

2. Figure 4c, what do the rows 'Metagenome' and 'Metagenomes' refer to? Some error in metadata curation?

3. Fosfomycin ARG (green in Figure 4C), seems quite high in Food Metagenomes, while it is barely present in panels A and B. I suppose this indicates uneven distribution of sampling 'host'? Also, is there known connection between food microbiomes and fosfomycin resistance? BTW, 'environment' will be a much better and accurate term than 'host'.

Minor comments:

1. Line 81: uploaded between 2010-01-01 and 2020-01-01 that had library source as 'METAGEOMIC'

2. Inconsistent line spacing starting on page 5 line 135 through page 6 line 152

3. Any data on how sequencing depth affects detection of ARGs or 16/18S genes? For example, Fig2a could be converted from a density plot to a scatterplot showing sequencing depth vs fragments

4. Figure 3, what do different colour shades mean for boxes?

Reviewer #3:

[Note: because of other commitments, this reviewer was only able to give us the following preliminary comments; we hope that they will nevertheless be useful]

My opinion on that paper is that it's a valuable analysis, and seems to have been done carefully. I had only two technical quibbles:

1. The manuscript does not explain how the analysis workflow handles two key issues. First: AMR databases are full of different versions of the same gene - e.g. there are more than 170 allelic versions of the CTX-M gene. Were all reads mapped to a database containing all of these, or were representatives chosen? If mapping to everything, what was done with reads that mapped to multiple alleles of one gene , and how were counts resolved? Two: I don't understand why, when calculating abundance, using counts of reads mapping to ribosomal RNA as a denominator makes sense, as rRNA arrays are different lengths in different species.

2. The text seems to suggest the same mapping workflow was used for nanopore, pacbio, and illumina. Is this really true? The same kmer size also? If yes, a lot of sensitivity will have been lost in the long read data , although since this is <10% of the data, this is not really a big issue.

I also had one red flag: Given the high rate of metadata errors in the ENA, I am suspicious of the samples dated between 1845 and 1905 in Figure 3 - is there a way to check these? If there is no associated publication discussing old metagenomes, I would honestly consider discarding those datapoints as mislabelled.

Figure 4 is great, v interesting!

---

## [Editor Report · Decision Letter 2]

9 Aug 2022

Dear Dr Martiny,

Thank you for the submission of your revised Methods and Resources "A curated data resource of 214K metagenomes for characterization of the global antimicrobial resistome" for publication in PLOS Biology. On behalf of my colleagues and the Academic Editor, Tobias Bollenbach, I'm pleased to say that we can in principle accept your manuscript for publication, provided you address any remaining formatting and reporting issues. These will be detailed in an email you should receive within 2-3 business days from our colleagues in the journal operations team; no action is required from you until then. Please note that we will not be able to formally accept your manuscript and schedule it for publication until you have completed any requested changes.

Sincerely,

Roli Roberts

Senior Editor

PLOS Biology

rroberts@plos.org